# Effect of Solid-Solid Phase Change Material's Direct Interaction on Physical and Rheological Properties of Asphalt

Haisheng Zhao [1,2], Jianmin Guo [3], Shijie Ma [2,*], Huan Zhang [2], Chunhua Su [2], Xiaoyan Wang [2], Zengguang Li [2], Jincheng Wei [2] and Shiping Cui [2]

1  School of Highway, Chang'an University, Xi'an 710064, China; zhaohaisheng@sdjtky.cn
2  Key Laboratory of Highway Maintain Technology Ministry of Communication, Jinan 250102, China; zhanghuan@sdjtky.cn (H.Z.); suchunhua@sdjtky.cn (C.S.); wangxiaoyan@sdjtky.cn (X.W.); lizengguang@sdjtky.cn (Z.L.); weijincheng@sdjtky.cn (J.W.); cuishiping@sdjtky.cn (S.C.)
3  Shandong Hi-Speed Company Limited, Jinan 250014, China; 18653160085@126.com
*  Correspondence: mashijie@sdjtky.cn; Tel.: +86-18660163082

**Abstract:** Asphalt pavement is a temperature−sensitive structure that is prone to temperature-related diseases. Phase change material (PCM) is an excellent candidate for mitigating these diseases. This paper looked into the effects of indirect composite shape-stabilized PCM incorporation on the characteristics of asphalt. The compatibility, physical properties, and rheological properties of asphalt with various PCM content before and after aging were thoroughly investigated. No phase separation and no chemical reaction occurred between PCM and asphalt. The physical properties improved with the addition of PCM, and the high−temperature performance indexes improved while the low−temperature performance indexes decreased as the aging process progressed. The effects of PCM on the rheological properties of the matrix and SBS−modified asphalt was distinct. PCM was added to improve the high−temperature rheological characteristics of the matrix asphalt when the temperature was higher than 52 °C, while PCM reduced the high−temperature rheological properties of the SBS−modified asphalt. The aging process has an impact on the high−temperature rutting factor of asphalt with a high PCM content. The low−temperature creep behavior and PG grade of asphalt were both improved. The implication of PCM is that it cannot increase the thermoregulation of asphalt pavement without the cost of scarifying the performance of the asphalt or mixture.

**Keywords:** phase change material; physical property; rheological property; aging; direct interaction





## 1. Introduction

Asphalt pavement is widely used in highway engineering as it can provide passengers with a high degree of comfort. Asphalt has low−temperature elasticity and high−temperature viscosity, which can be attributed to its time and temperature sensitivity properties [1–4]. Thermal fatigue cracking, low−temperature cracking, and freeze−thaw cycling have a significant impact on the service performance and service life of asphalt pavement [5]. Repeated temperature cycles are the main trigger for the temperature sensitivity of asphalt, high tensile stress, weakened tensile strength, and repeated high/low stress cycling produced by temperature fluctuations and loading [6]. Thus, significant research efforts have focused on developing technologies and materials to prevent temperature defects in asphalt pavement [7–10].

Phase change energy storage technology, which has been proven to effectively reduce the magnitude of extreme temperature variations and extend the asphalt pavement service life, is one potential approach to solving the aforementioned temperature−related damages. Phase change material (PCM) can change its phase state, e.g., solid to solid, solid to liquid, solid to gas, liquid to gas, or vice−versa [11–13]. PCM has a small volume change throughout the phase change process and essentially keeps constant temperature by absorbing or transmitting energy (often known as latent heat) to the surrounding environment during

melting or crystallization [14,15]. Phase change only depends on temperature; therefore, PCM is an excellent candidate for the thermoregulation of asphalt pavement and the reduction of temperature−related damages. PCM has various advantages, including high energy storage density, approximate isothermal process, a broad selection range of phase transition temperature, easy to control, and large heat−storage capacity [14,16].

The PCM is divided into different groupings based on the nature of the materials, such as organic, inorganic, and eutectic. Fatty acids, glycerine, phenol, caprylone, and other non−paraffin PCM subgroups are among the organic materials that are further classified as paraffin and non−paraffin. The inorganic materials are further classified as salt hydrate and metallics.

Direct incorporation [17], micro−encapsulate, macro−encapsulate [18,19], and aggregate impregnation methods are some of the PCM application methods that could be employed as modifiers for asphalt or fillers for asphalt mixtures. Each of these approaches has its advantages, as well as obvious limitations. Although direct incorporation is easy to carry out, the chemical compositions change significantly [17,19–22]. Although the micro−encapsulation method could protect PCM leaks from micro−capsules, the heat−transmission efficiency is insufficient to meet the requirements of temperature regulation of asphalt pavement [23–25]. The aggregate impregnation method could provide adequate protection from leakage, but its absorption capacities and temperature regulation performance are limited [26–29]. Challenges and solutions associated with the incorporation of PCM in asphalt or asphalt mixtures must be considered, and more research is required to find a reasonable approach to incorporate PCM in asphalt or asphalt mixtures.

Direct contact with asphalt or PCM leakage from micro−encapsulation would increase the aromatic and saturate fractions of asphalt and affect the colloidal structure of the asphalt binder [17,22], which would be of no benefit for storing or releasing energy in asphalts. The direct interaction of PCM with asphalt, as well as its impact on the physical and rheological properties of asphalt, must be investigated. Kakar et al. [30] studied the effect of asphalt's direct interaction with tetradecane on the properties of asphalt and found that the thermal energy released by PCM crystallization during cooling improved the rheological properties. Du et al. [31] investigated the effect of polyethylene glycol on direct interactions with asphalt. Tan et al. [22] found that the physical properties of asphalt that had been directly affected by organic phase change materials had been seriously impacted. The properties of asphalt modified with organic PCM cannot match the standards [32]. Zhang et al. [15] demonstrated that PCM enhances the rutting resistance of asphalt at high temperatures while also increasing the risk of fatigue and cracking.

Shape−stabilized technology, such as shape−stabilization [15], micro−encapsulation [33], and macro−encapsulation [34] is used to create composite PCM, which could withstand (a) the external force during mixing, paving, and compacting activities of asphalt pavement, and (b) high temperature to prevent PCM leakage. Zhang et al. [35] fabricated shape−stabilized composite PCM to conquer PCM leakage. Ma et al. [36] used an organic phase separation model to generate composite−modified PCMs with ethyl cellulose as the wall material and modified PCM as the core material. Composite shape−stabilized PCM was blended in asphalt mixtures to prevent the direct interaction of PCM with asphalt. Solid−solid PCM show clear advantages over solid−liquid PCM, including no liquid leakage problem, adjustable temperature region, good thermal stability, excellent durability, and high storage density [37–40].

Most research has focused on the application of PCM in asphalt mixture, with little attention paid to the impact of PCM on the properties of the asphalt binder. Wei [41,42], Zhang [15], Bueno [43] investigated the physical properties, rheological properties, and temperature adjustable performance of asphalt modified with polyurethane solid−solid PCM, expanded graphite (EG)/polyethylene glycol (PEG) composite PCM, and micro−encapsulated tetradecane PCM, respectively.

This paper deals with the influence of solid−solid composite shape−stabilized PCM, which are directly added to asphalt, on the properties of matrix and SBS−modified asphalt.

The investigated properties include physical properties (e.g., penetration, softening point, ductility, rotational viscosity) before and after aging, and rheological properties (e.g., high− and low−temperature rheological properties) before and after aging. Moreover, the morphology and reaction between PCM and asphalt will also be examined. The feasibility of the method of directly incorporating solid–solid composite shape−stabilized PCM in asphalt should be verified. This work will provide an insight into how solid−solid PCM affects the physical and rheological properties of asphalt before and after aging and attempts to attain a selection principle for the implication of PCM materials in asphalt pavement.

## 2. Materials and Methods

### 2.1. Materials

Matrix asphalt and SBS−modified asphalt were tested in this paper; the physical properties are listed in Table 1.

**Table 1.** The physical parameters matrix and SBS−modified asphalt.

| Index | Matrix Asphalt | SBS−Modified Asphalt |
|---|---|---|
| Penetration (0.1 mm) | 69.4 | 65.9 |
| Softening Point (°C) | 51.5 | 46.7 |
| Ductility (cm) | 64 | 93 |
| Rotary Viscosity (135 °C, MPa·s) | 0.881 | 2.797 |

Solid−solid composite shape−stabilized phase change materials (namely PCM) used in this paper were provided by a commercial company, the physical and chemical parameters of PCM are listed in Table 2.

**Table 2.** The physical and chemical parameters of solid−solid phase change material (PCM).

| Parameter | Test Result |
|---|---|
| Apparent density (kg/m$^3$) | 840 |
| Latent heat value (Melting enthalpy value) (J/g) | 67 |
| Phase change point (°C) | 17 |
| Phase change interval (°C) | 0–36 |
| Phase change type | solid−solid |
| The decay rate of latent heat value (Melting enthalpy value) after 20 times phase change (%) | 2 |

To investigate the influence of PCM on the physical and rheological properties of matrix and SBS−modified asphalt, the PCM was added to the matrix and SBS−modified, and the concentration of PCM added was 4% and 8% by the weight of the asphalt. Before testing, different contents of PCM were blended with asphalt in the laboratory. The blending processes were conducted using a high−speed mixer at 3000 rpm for 60 min. For matrix and SBS−modified asphalt, the blending temperatures were 140 °C and 170 °C, respectively.

### 2.2. Methods

#### 2.2.1. Physical Property Test

Before modification, the physical properties of the control pure asphalt binders were tested to provide a basis to understand the impacts of PCM on asphalt binder properties.

The physical properties (e.g., penetration degree, softening point, ductility, rotational viscosity) of asphalts with various contents of PCM were tested according to the requirements of the Standard Test Methods of Bitumen and Bituminous Mixtures for Highway Engineering (JTG E20−2011) T0604, T0605, T0606, T0620, T0625 [44]. The physical properties are related to the high−temperature and low−temperature performance of asphalt.

### 2.2.2. Pressurized Aging Vessel (PAV) Test

Unmodified and PCM modified asphalts were subjected to the pressurized aging vessel (PAV) test following the Standard Test Methods of Bitumen and Bituminous Mixtures for Highway Engineering (JTG E20−2011) T0630 [44] to verify the influence of aging. The physical properties of asphalts were examined after artificial aging procedures.

### 2.2.3. Fluorescence Microscope

The phase composition and morphology were important for confirming the fusion status of the asphalt and PCM. The fluorescence microscope (LW2000C−LY, CEWEI, Shanghai, China) was employed in this investigation to observe the distribution of PCM in the asphalt immediately after modification.

### 2.2.4. Fourier Transform Infrared Spectroscopy (FTIR)

Fourier transform infrared spectroscopy (FTIR, Bruker TENSO II, Bruker, Karlsruhe, Germany) with wavenumbers ranging from 400 to 4000 cm$^{-1}$ and KBr as the dispersing phase was used in this paper, the functional groups of the control asphalts and PCM−modified asphalts were analyzed to verify whether the PCM reacts with asphalt.

### 2.2.5. Rheological Property Tests

Dynamic shear rheometers (DSR, TA, New Castle, DE, USA) were used to examine the high−temperature rheological properties of the control asphalts and asphalts with various contents of PCM. The temperature sweep in DSR was measured in the strain−controlled mode at a constant frequency (10 rad/s). The frequency sweep test was performed at a constant strain of 9%. For the matrix asphalt, test temperatures ranged from 40 °C to 76 °C with 12 °C intervals. For the SBS−modified asphalt, temperatures ranged from 52 °C to 88 °C with 12 °C intervals. The frequency range was from 0.1–10 Hz, which corresponded to the traditional vehicle speed. To simulate short−term aging, the asphalt samples used in the DSR test were artificially aged by using the Rolling Thin Film Oven test (RTFOT).

The creep behaviors of the asphalt samples at the low temperature were investigated using a bending beam rheometer (BBR, CANNON, Centre County, PA, USA) test, with temperatures of −6 °C, −12 °C, and −18 °C. The Rolling Thin Film Oven test (RTFOT) was utilized to artificially age the asphalt samples used in the BBR test, and the RTFOT residues were subjected to the pressure aging vessel (PAV) to simulate the long−term aging of the asphalt samples.

The Standard Test Methods of Bitumen and Bituminous Mixtures for Highway Engineering (JTG E20−2011) T0628, T0627 [44] were used to conduct the rheological property testing.

## 3. Results

### 3.1. Physical Property Test Results

The effects of PCM on asphalt penetration, softening point, ductility, and rotary viscosity are shown in Figure 1.

As seen in Figure 1a, the penetration at 25 °C increased after the addition of PCM, but gradually reduced as the PCM content increased. When the PCM content was 4 wt% and 8 wt%, penetration values increased for the matrix asphalt by 16.1% and 12.2% and for the SBS−modified asphalt by 27.6% and −4.6%, respectively. The addition of PCM had a greater effect on the penetration of SBS−modified asphalt than matrix asphalt.

An asphalt binder with a lower penetration degree is more resistant to rutting in high temperatures. Comparing the penetration values in Figure 1a, all combinations had a higher penetration degree except SBS−modified asphalt with 8 wt% PCM. Furthermore, 4 wt% PCM had a stronger effect on increasing the penetration degree than 8 wt%. The addition of PCM could weaken the matrix asphalt's high−temperature resistance.

It can be noted from Figure 1b that with the addition of PCM, the softening point of matrix asphalts reduced slightly but increased as the PCM content increased. The softening

point increased by 5.7 °C. For SBS−modified asphalt, the softening point increased by 7.3 °C.

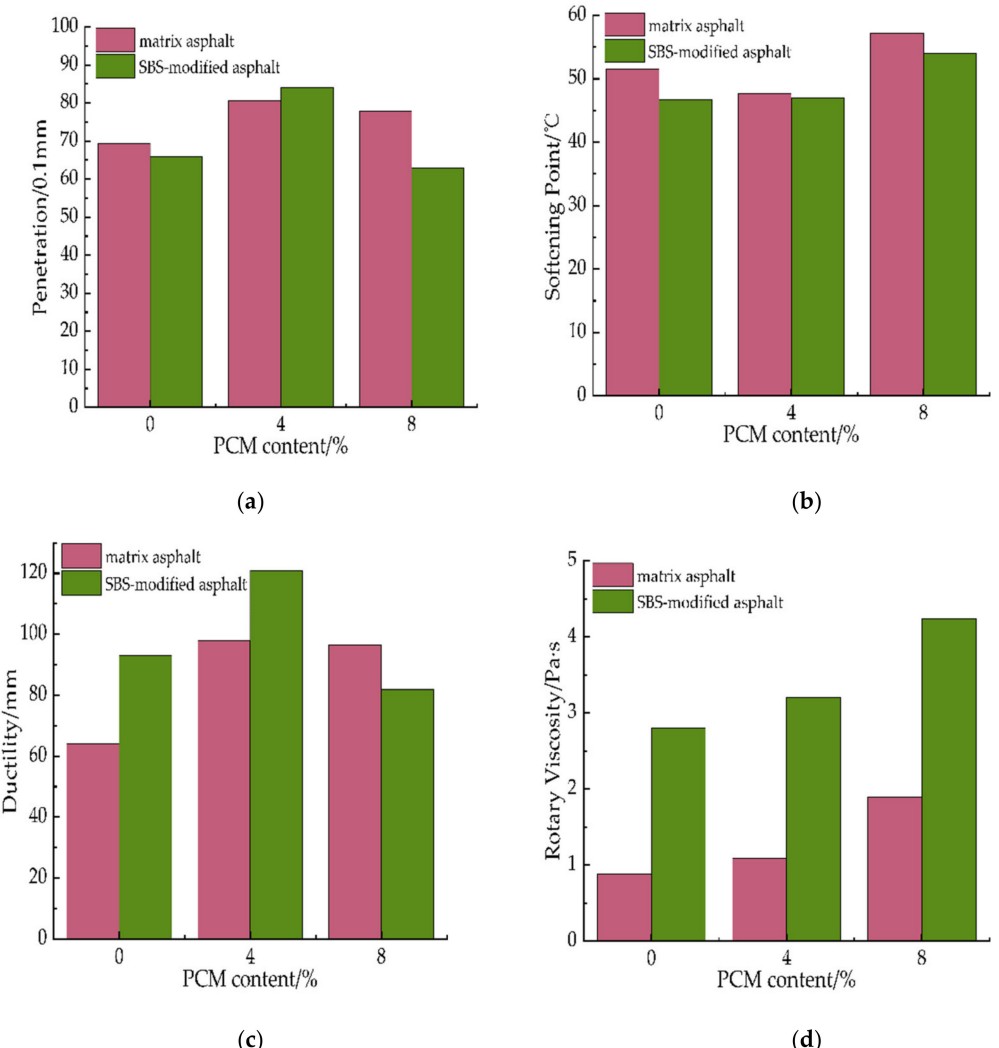

**Figure 1.** Physical property test results of asphalt samples with and without PCM. (**a**) penetration test results; (**b**) softening point test results; (**c**) ductility test results; (**d**) rotary viscosity test results.

The effect of PCM on the ductility was similar to the penetration index, as illustrated in Figure 1c. When the PCM addition was 4 wt%, the ductility values of the matrix asphalt (at 10 °C) and SBS−modified asphalt (at 5 °C) increased by 53.1% and 30.1%; when PCM was 8 wt%, the ductility values increased by 50.8% and −11.8%. Therefore, the high content of PCM reduces the cracking resistance of SBS−modified asphalt at low temperatures.

Figure 1d shows the rotary viscosity values of matrix and SBS−modified asphalt samples at 135 °C. As the PCM concentration increased, the rotary viscosity increased steadily. For 4 wt% and 8 wt% PCM, the increasing ratio of matrix asphalt was 23.1% and 114.5%, and for the SBS−modified asphalt, the increasing ratio was 14.4% and 51.4%, respectively.

## 3.2. Physical Property Test Results after Ageing

Penetration, softening point, ductility, and mass loss tests were performed to analyze the changes in the conventional physical properties of asphalts before and after PAV aging, the results are plotted in Figure 2.

During the asphalt pavement service period, the characteristics of asphalt vary with time and temperature (asphalt aging). Short−term aging during mixing and storage, as well as long−term aging throughout service, are both simulated at the laboratory scale

employing thermal and oxidative treatment. In this study, the RTFOT residues were subjected to a pressure aging vessel (PAV) to simulate the long−term aging of asphalt.

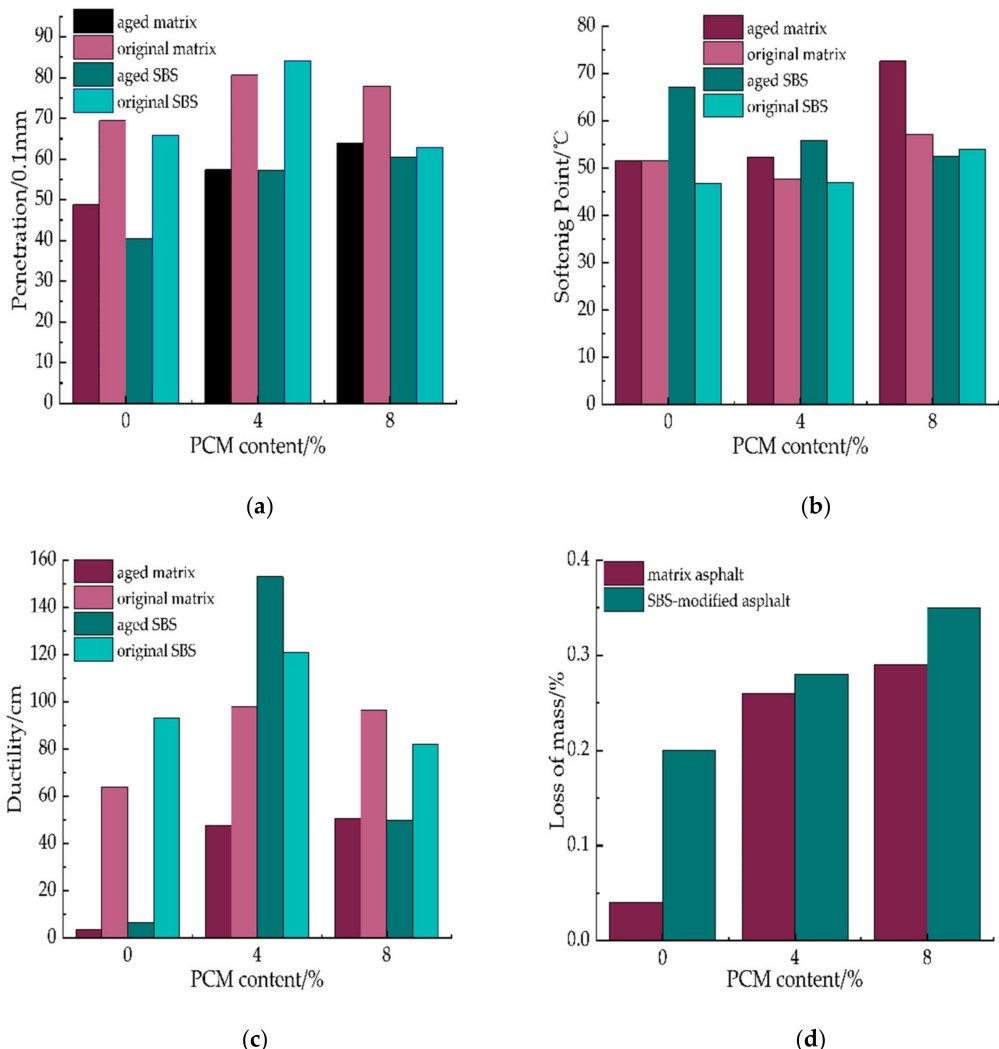

**Figure 2.** Physical property test results of asphalt samples before and after PAV aging. (**a**) penetration test results; (**b**) softening point test results; (**c**) ductility test results; (**d**) loss of mass test results.

Figure 2a illustrates how the penetration values of the asphalt samples before and after modification reduced after the PAV aging process. The decreasing trend in penetration values before and after PAV aging was reduced with the addition of PCM to asphalt, implying that adding PCM could postpone the effect of aging on the penetration index.

The softening point of asphalt with various contents of PCM before and after PAV aging increased, except for SBS−modified asphalt with 8% PCM, as shown in Figure 2b. However, the effect ratios for matrix and SBS−modified asphalt with varied PCM concentrations were incompatible. As PCM content grew, the difference between the matrix asphalt samples' softening points before and after PAV aging increased.

From Figure 2c, it can be noticed that, except for SBS−modified asphalt with 4 wt% PCM, asphalt samples before PAV aging had higher ductility. In comparison to the original asphalt samples, adding PCM potentially improved the cracking resistance after aging.

The loss of mass values after the PAV aging grew following the PCM content increasing, as shown in Figure 2d. The growth was not substantial, indicating that asphalts mixed with PCM had good thermal stability.

*3.3. Morphology of the Virgin Asphalt and PCM-Modified Asphalt*

Figure 3 displayed fluorescence images (magnified by 400 times) of asphalt samples with varying PCM content. Fluorescence images were used to investigate the distribution of PCM in asphalt.

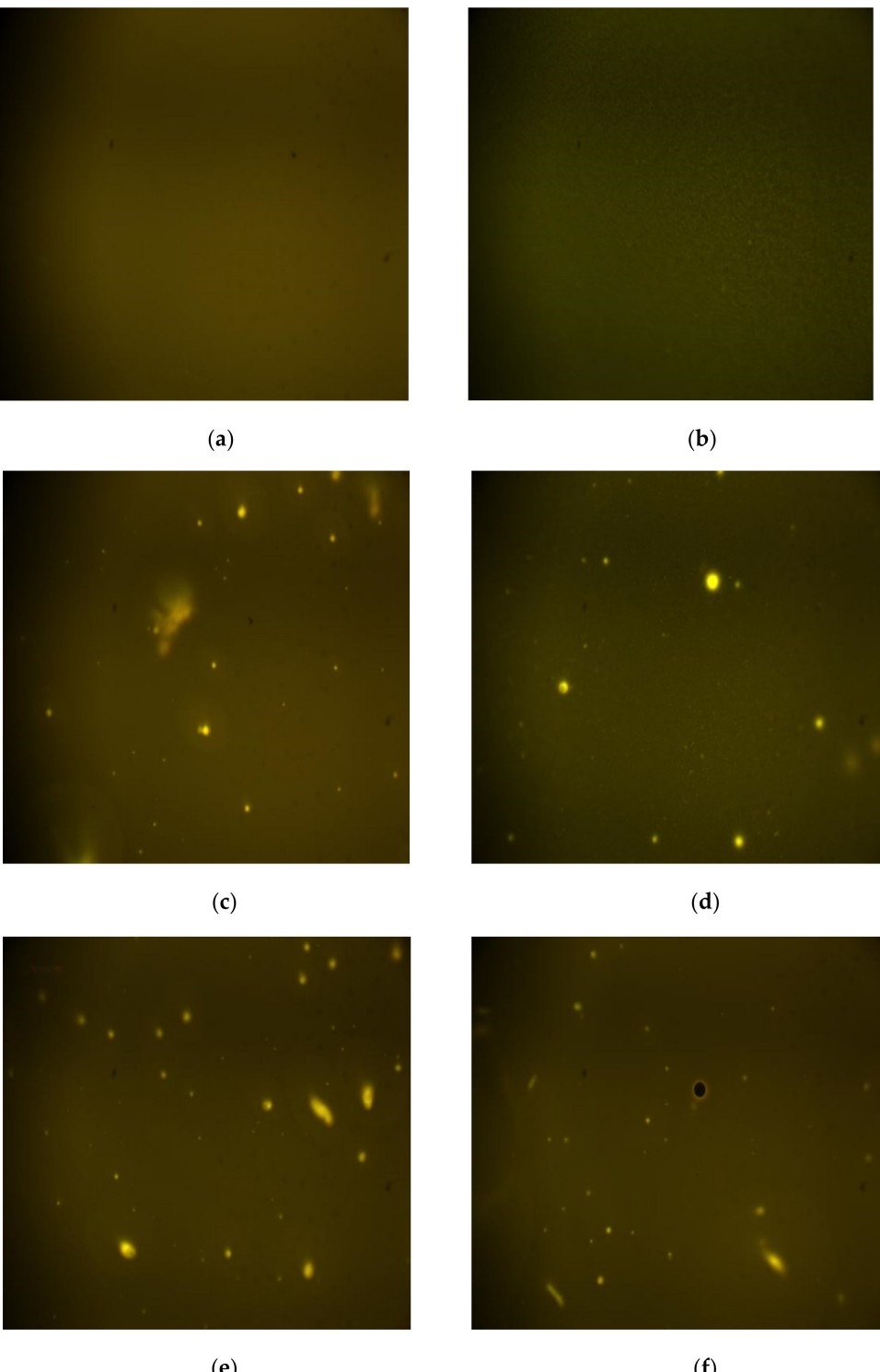

**Figure 3.** The morphology of asphalt samples before and after modification (magnified by 400 times). (**a**) pure matrix asphalt; (**b**) pure SBS−modified asphalt; (**c**) matrix asphalt with 4 wt% PCM; (**d**) SBS−modified asphalt with 4 wt% PCM; (**e**) matrix asphalt with 8 wt% PCM; (**f**) SBS−modified asphalt with 8 wt% PCM.

The virgin asphalt exhibits a flat and smooth fracture surface with no gully or layer structure, as illustrated in Figure 3a,b. It can be found that PCM was randomly distributed in the form of very small particles and these particles were always separated. When the PEG content increased, the particle number was multiplied. The phase interface between PCM particles and asphalt indicated that the PCM particles did not react with the asphalt.

### 3.4. FTIR Analysis

The FTIR (Bruker TENSO II, Bruker, Karlsruhe, Germany) analysis was used to determine the chemical changes in the asphalt samples. Each specific chemical bond can be defined by its specific wave−number range ($cm^{-1}$).

The FTIR spectra of matrix and SBS−modified asphalt samples before and after modification are depicted in Figure 4. The infrared spectrum of asphalt samples with different PCM content was highly similar to that of pure asphalt. In the spectra of asphalt samples with 4 wt% and 8 wt% of PCM, a new absorption peak at 1737 $cm^{-1}$ (−C=O functional group) was found. The absorption peak increased as the PCM content increased, implying that the absorption peak occurred as a result of the addition of PCM. It could be attributed to the residual PCM that remained on the surface after the PCM microcapsules were prepared.

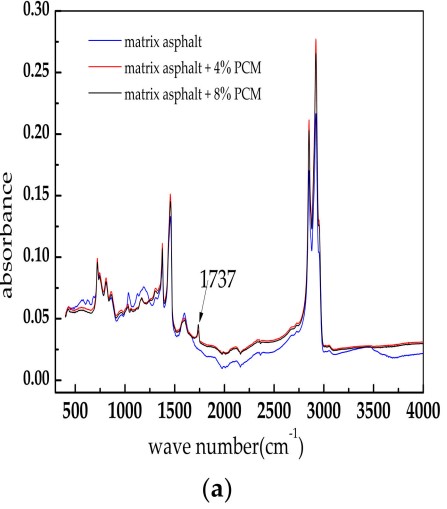
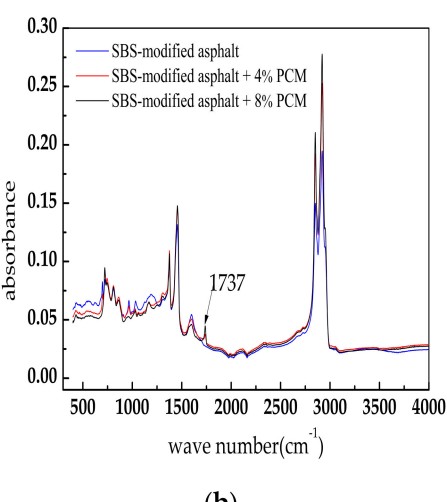

(**a**) (**b**)

**Figure 4.** FTIR spectra of asphalt samples before and after modification. (**a**) matrix asphalt samples; (**b**) pure SBS−modified asphalt samples.

### 3.5. Effect of Aging and PCM Modification on Rheological Properties of Asphalt

3.5.1. High−Temperature Rheological Properties

High−Temperature Rheological Properties of Original Asphalt Samples

The frequency sweep test, the complex shear modulus, phase angle, and corresponding rutting factor are plotted in Figure 5.

It is shown in Figure 5a, that the complex shear modulus of the unaged matrix asphalt samples increased as the loading frequencies increased. The addition of PCM did not affect this trend. Under 40 °C, the complex shear modulus of the matrix asphalt with no PCM was the largest. When the test temperature was raised to 52 °C, the complex shear modulus of the matrix asphalt with no PCM and 8 wt% PCM was close and the complex shear modulus at 4 wt% was the smallest. The matrix asphalt with 8 wt% PCM had a higher complex shear modulus when the test temperature was raised to 64 °C and 76 °C. The differences between the three kinds of matrix asphalt samples grew as the test temperature was raised.

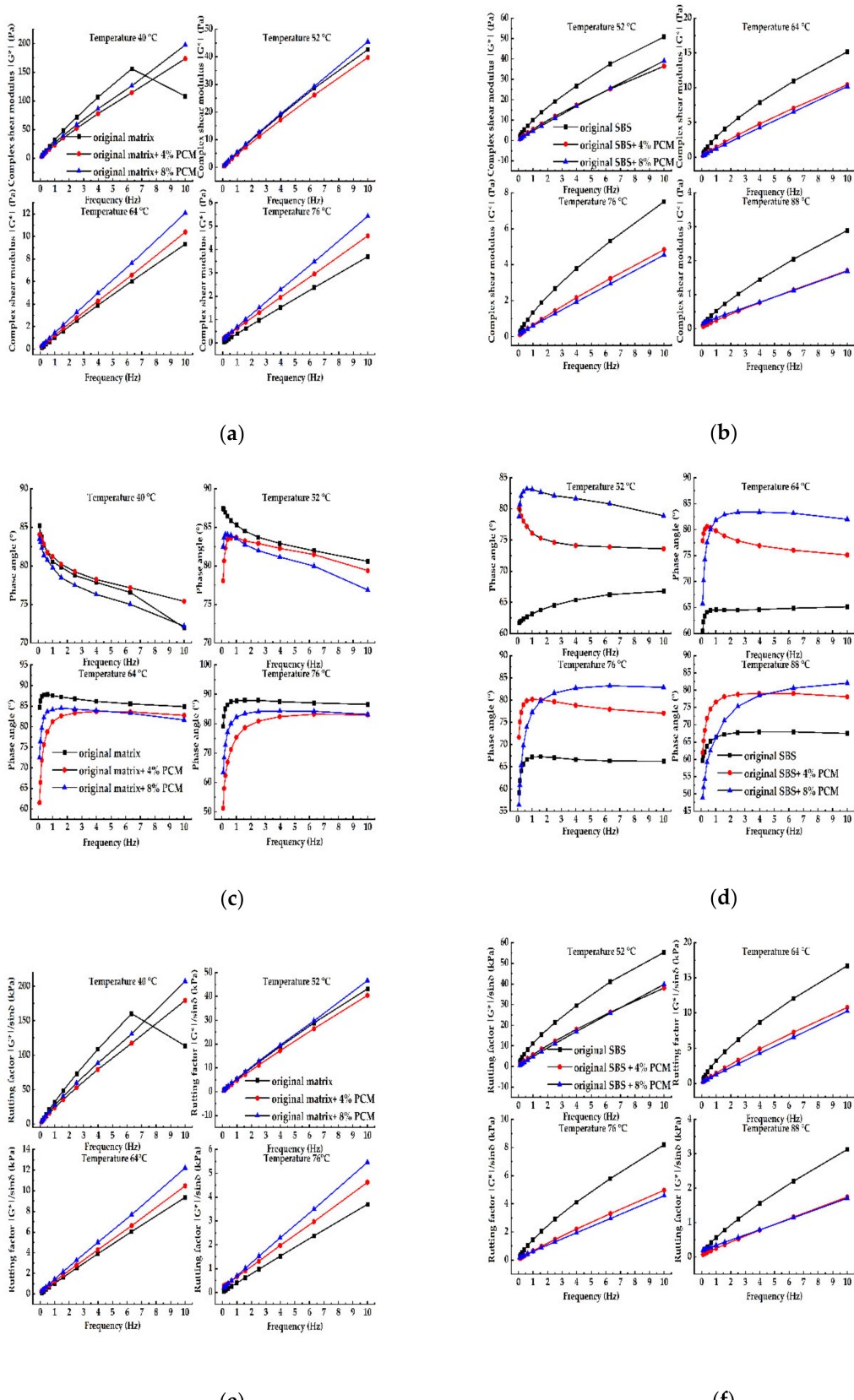

**Figure 5.** DSR frequency sweep test results of original asphalt samples with PCM under different test temperatures. (**a**) complex modulus of matrix asphalt samples; (**b**) complex modulus of SBS−modified asphalt samples; (**c**) phase angle of matrix asphalt samples; (**d**) phase angle of SBS−modified asphalt samples; (**e**) rutting factor of matrix asphalt samples; (**f**) rutting factor of SBS−modified asphalt samples.

The matrix asphalt without PCM had the highest complex shear modulus when the test temperature was lower. With the test temperature raised, the effect of PCM on the complex shear modulus became more significant, indicating that the solid−solid PCM started to absorb the heat. The inner heat was absorbed by the added PCM, which enhanced the complex shear modulus. The amount of PCM added had a significant impact on the complex shear modulus.

It can also be noted, from Figure 5b, that the complex shear modulus of SBS−modified asphalt becomes bigger with PCM added. The amount of PCM added did not affect the complex shear modulus of the matrix asphalt with the raising of the test temperature.

According to Figure 5c, the phase angle of the matrix asphalt with various PCM concentrations increased with the raising of the test temperature. At low loading frequencies, the phase angle of the matrix asphalt with PCM rose dramatically.

The effect of adding PCM on the phase angle of the SBS−modified asphalt was more complex than that of the SBS−modified asphalt without PCM, as shown in Figure 5d.

In comparison to the complex shear modulus, the rutting factors of the original matrix and SBS−modified asphalt with varied PCM content exhibited comparable trends, as seen in Figure 5e,f.

High−Temperature Rheological Properties of Aged Asphalt Samples

Figure 6a shows that the complex shear modulus of the aged matrix asphalt with 8 wt% PCM was the highest. The difference between the complex shear modulus of the aged matrix asphalt with 4 wt% and with 0 wt% PCM becomes insignificant with the test temperature raised. While the gap of the complex shear modulus between the aged matrix asphalt with 8 wt% PCM and other asphalts grows. The complex shear modulus of the aged matrix asphalt with various PCM concentrations was significantly higher than the unaged.

As can be observed from Figure 6b, under test temperatures of 52 °C and 64 °C, the complex shear modulus difference between aged the SBS−modified asphalt with 8 wt% and with 0 wt% PCM was insignificant. However, when the test temperature was raised to 76 °C and 88 °C, the complex shear modulus of the aged SBS−modified asphalt without PCM became the highest. The complex shear modulus of the aged SBS−modified asphalt with various PCM concentrations was significantly higher than the unaged.

The RTFOT aging process had a greater impact on the phase angle of the matrix and SBS−modified asphalt with 8 wt% PCM, as seen in Figure 6c,d.

In comparison to the complex shear modulus, the rutting factors of the aged matrix and SBS−modified asphalt with varied PCM content exhibited similar trends. Rutting factors under all test temperatures were bigger than that of original asphalt samples, as shown in Figure 6e,f.

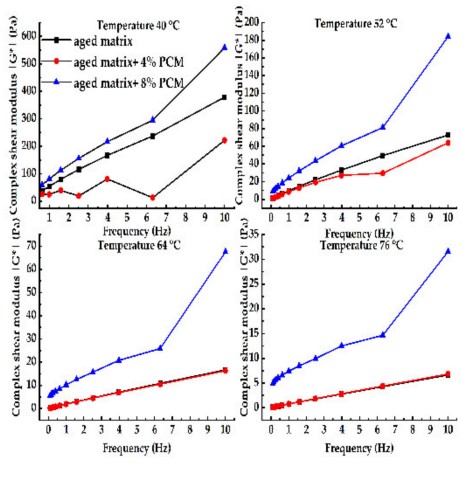　　　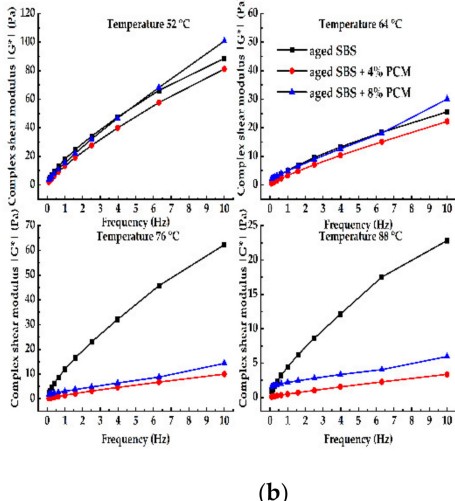

(a)　　　　　　　　　　　　　　　　　　　　　(b)

**Figure 6.** *Cont.*

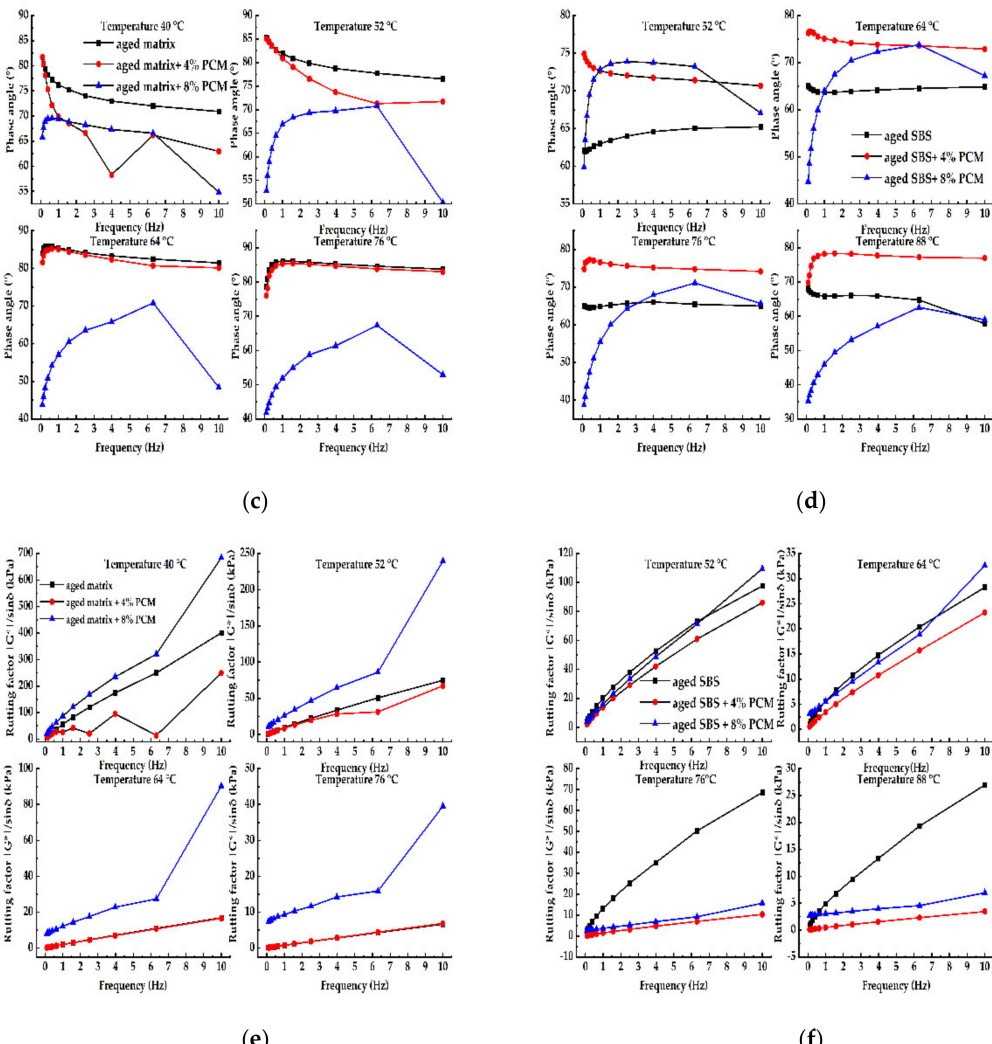

**Figure 6.** DSR frequency sweep test results of RTFOT aged asphalt samples with PCM under different test temperatures. (**a**) complex modulus of matrix asphalt samples; (**b**) complex modulus of SBS−modified asphalt samples; (**c**) phase angle of matrix asphalt samples; (**d**) phase angle of SBS−modified asphalt samples; (**e**) rutting factor of matrix asphalt samples; (**f**) rutting factor of SBS−modified asphalt samples.

### 3.5.2. Low−Temperature Rheological Properties

Figure 7 depicts the effect of PCM on the creep stiffness (donated as St) and creep curve slope (donated as m−value) of the matrix and SBS−modified asphalt with different PCM content.

It can be seen from Figure 7a, that the St of the matrix asphalt decreased as the PCM content and test temperature increased. The addition of PCM would improve the matrix asphalt's low−temperature crack resistance.

Adding PCM to the SBS−modified asphalt reduced St at all test temperatures, with 4 wt% PCM having the greatest effect, as shown in Figure 7b. Low−temperature resistance was reduced in the SBS−modified asphalt with PCM.

It can be observed, from Figure 7c,d, that the increase in PCM did not affect the m−value. Only 4 wt% PCM could increase the m−value when compared to asphalts without PCM.

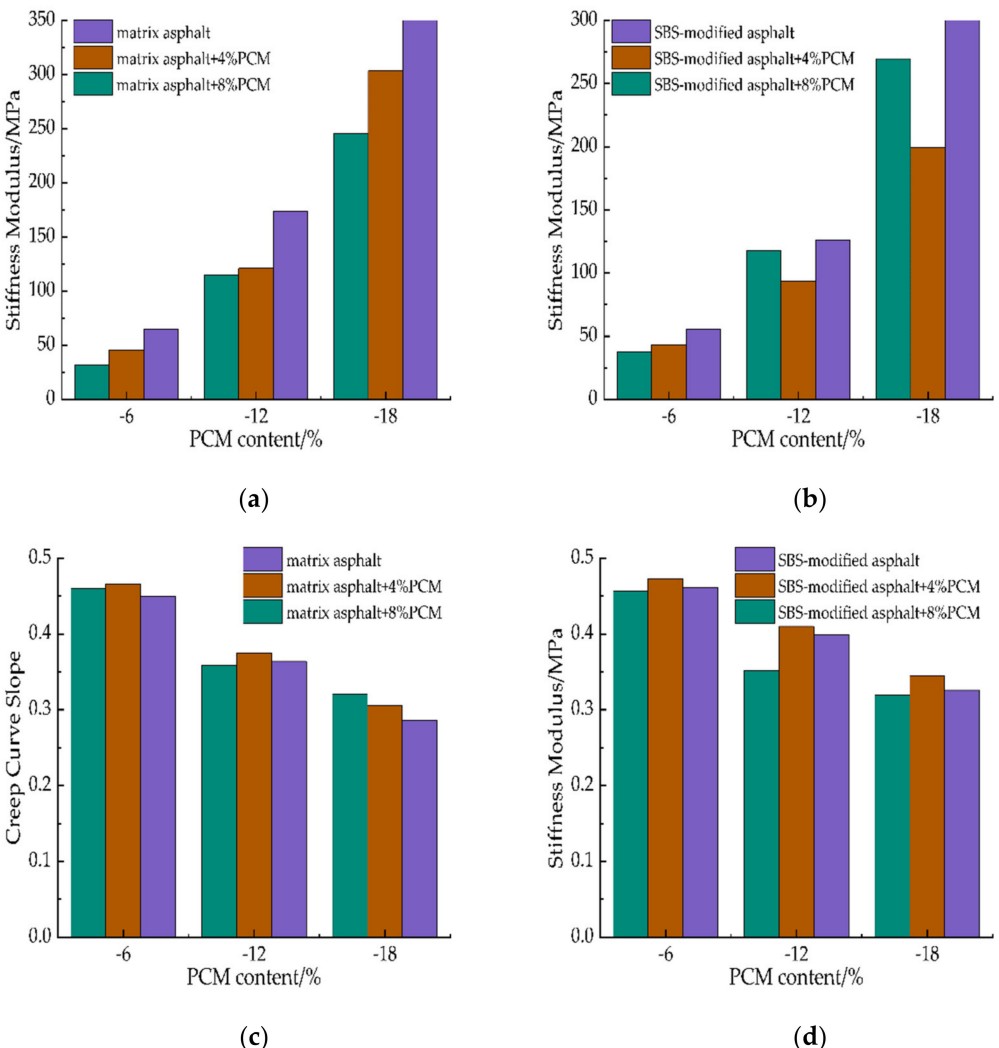

**Figure 7.** BBR test results of asphalt samples with PCM under different test temperatures. (**a**) stiffness modulus of matrix asphalt samples; (**b**) stiffness modulus of SBS−modified asphalt samples; (**c**) creep curve slope of matrix asphalt samples; (**d**) creep curve slope of SBS−modified asphalt samples.

## 4. Discussion

### 4.1. Effects of PCM Modification on Physical Properties of Asphalt

Previous research suggests that the incorporation of PCM in asphalt is not a feasible method for PCM application [22]. The impact of PCM on asphalt must be investigated as PCM particles would inevitably have contact with the asphalt binder during mixing and paving activities.

At high temperatures, asphalt with a lower penetration degree or a greater softening point is more resistant to rutting. Considering the penetration and softening point values in Figure 1, the rutting resistance at high temperatures was reduced with 4 wt% PCM. The effects of 8 wt% PCM on penetration and softening point values were adverse, indicating that the addition of PCM at different contents had a complex impact on the sensitivity to high temperature when compared to neat asphalt. Kakar et al. [30] indicate that the direct integration of PCM (tetradecane, Tm 6C) enhances the penetration, which is consistent with the results of Figure 1a. This issue could be due to the leakage of PCM microcapsules. During the preparation of PCM−modified asphalt, a limited number of micro−capsules may break due to high temperature and external force, thereby weakening the bitumen binder and increasing penetration.

The PCM went through a phase change during the heating process, which resulted in a higher softening point. These findings were in line with what was expected [42]. However, the softening point was slightly enhanced due to the effective micro−capsule protection against PCM.

The asphalt with higher ductility is more resistant to cracking at low temperatures. All combinations, except for the SBS−modified asphalt with 8 wt% PCM, were less sensitive to low temperatures than the neat asphalt binders in terms of ductility.

The workability of HMA is highly dependent on the viscosity of asphalt at high temperatures. Regarding the rotary viscosity values in Figure 1, the viscosity of PCM−modified asphalt rose as the PCM content increased. Furthermore, the rotary viscosity values of the matrix asphalt modified with PCM were smaller than the maximum acceptable (3 Pa·s). The rotary viscosity values of the SBS−modified asphalt mixed with various concentrations of PCM exceeded the maximum acceptable (3 Pa·s). Moreover, mixing PCM with SBS−modified asphalt increases asphalt viscosity, which indicates that mixing and compaction temperatures should be increased. It is a challenge to combine PCM with SBS−modified asphalt.

### 4.2. Aging of PCM−Modified Asphalt

Because of time, temperature, and ultraviolet rays, the asphalt binder ages during service life, resulting in changes in the physical behavior (e.g., stiffening) of asphalt pavement. The asphalt aging process is simulated in the laboratory for (a) short−term aging during mixing and storage; (b) the long−term aging during service life by applying thermal and oxidative treatment. It can be seen that adding PCM to asphalt reduces penetration, increases the softening point (except in SBS−modified asphalt with 8 wt% PCM), decreases ductility (except in SBS−modified asphalt with 4 wt% PCM), and increases the loss of mass.

When compared to the unaged asphalt samples, the high−temperature performance of the matrix and SBS−modified asphalt with PCM added was improved, while the low−temperature performance was lowered. In conclusion, adding PCM to asphalt is beneficial to the stability of asphalt at high temperatures, as well as its aging resistance and durability.

### 4.3. Morphology of the Original Asphalt and PCM−Modified Asphalt

It is noteworthy that the PCM was blended with asphalt in the form of small particles based on the fluorescence image results of the matrix and SBS−modified asphalt with varying PCM concentrations. PCM particles were visible, as was the interface between PCM and asphalt, indicating poor compatibility. When the PCM content was 4 wt%, phase separation emerged. The poor compatibility of PCM and asphalt would damage the durability and reliability of as−prepared blends, as well as prevent the use of higher PCM content.

When the PCM content reached 8 wt%, the phase separation between PCM and SBS−modified asphalt became lower than that between PCM and matrix asphalt (see Figure 3). This phenomenon could be explained by the styrene−butadiene−styrene network generated in the SBS−modified asphalt, which can settle the PCM particles, and so prevent phase segregation to some extent.

According to the test results, there was an obvious phase interface between PCM particles and asphalt. The PCM particles were in a dispersed phase. There was no dissolution reaction between PCM particles and asphalt. The spatial mesh structure of the co−blending system in the SBS−modified asphalt was not affected by the addition of PCM.

### 4.4. FTIR Analysis

In the spectra of asphalts modified with PCM, only a new absorption peak at 1737 cm$^{-1}$ belonging to the stretching vibration peaks of the −C=O functional group was discovered. According to FTIR analysis, other peaks revealed the specific absorption characteristic of the matrix and SBS−modified asphalt. This new peak was attributed to the typical absorption

of residual PCM. It can be concluded that no chemical reactions occurred between the asphalt binders and additives. The PCM was physically integrated into the asphalt without causing a chemical reaction.

*4.5. Effect of Aging and PCM Modification on Rheological Properties of Asphalt*

4.5.1. High−Temperature Rheological Properties

Frequency sweep tests were performed at four different test temperatures to evaluate the PCM phase state effect on the viscosity components of the asphalt. High values of complex shear modulus (G*) and low values of phase angle (δ) are desirable for improving asphalt fatigue resistance. Asphalts with a bigger phase angle (δ) are more sensitive to viscous deformation than asphalts with the same complex shear modulus (G*). According to Superpave specifications, G*/sin(δ) is a rutting factor that represents a measure of asphalt binder rutting resistance [45].

According to the results obtained from the high−temperature rheological properties analysis, the rheological measurements show a combined effect of the steep drop in complex modulus (G*) and increase in phase angle (δ) with temperature change. The phase angle (δ) did not increase consistently with the decrease in complex shear modulus (G*). At the high temperature, the phase angle (δ) would grow at low frequency and then reduce at high frequency. At high temperatures, the phase angle (δ) did not change much. This is a complicated topic that is partly explained by the phase state of PCM and its impact on the rheological properties of asphalt.

The complex shear modulus (G*) of the matrix asphalt modified with PCM reduced as the PCM content increased in the whole sweep frequency when the test temperature was less than 52 °C; when the test temperature was greater than 52 °C, the trend was reversed. As a result, integrating PCM into asphalt was beneficial for its complex shear modulus (G*) at high temperatures. Furthermore, the rutting factor (G*/sin(δ)) of the matrix asphalt with various PCM content exhibited remarkably comparable variation trends with the complex shear modulus (G*), implying that PCM may improve the high−temperature deformation resistance of matrix asphalt. Adding PCM to SBS−modified asphalt did not improve its rutting resistance under all contents and temperatures. SBS−modified with varied PCM content showed no phase change behavior at test temperatures above 52 °C.

The RTFOT aging process altered the rheological properties of asphalt with PCM added. The rutting factor (G*/sin(δ)) of asphalt with variable PCM concentrations was higher after aging than that of unaged, indicating that the aging process may improve the resisting deformation at high temperatures.

After aging, the matrix asphalt with 8 wt% PCM had the highest complex shear modulus (G*) under all test temperatures and loading frequency. As the temperature rose, the complex shear modulus (G*) of asphalt with 4 wt% and no PCM are simliar. At 52 °C and 64 °C, the complex shear modulus (G*) and rutting factor (G*/sin(δ)) of aged SBS−modified asphalt with 8 wt% and no PCM were close and slightly higher than those of aged SBS−modified asphalt with 4 wt% PCM. The complex shear modulus (G*) and rutting factor (G*/sin(δ)) of aged SBS−modified asphalt with PCM added decreased significantly with an increase in temperature and were subsequently smaller than those of SBS−modified asphalt without PCM. According to the rheological analysis of aged asphalt, the high content of PCM improves matrix asphalt's high−temperature deformation resistance. Adding PCM could reduce rutting resistance at high temperatures.

By combining variations of the complex shear modulus (G*) and phase angle (δ) at different temperatures, it is difficult to explicitly present the influence of PCM addition on the high−temperature rheological property of asphalt.

4.5.2. Low−Temperature Rheological Properties

BBR tests were conducted at low temperatures to analyze the creep behavior of asphalts with various PCM concentrations. The St and m−value indices were used in BBR testing to assess creep behavior. The ability of asphalt to withstand a constant load is

measured in St, and asphalts with high St values are more brittle and prone to cracking. The m−value depicts how the stiffness of the asphalt changes as the loads are applied. The binder stiffness changes quickly with a high m−value, the accumulated thermal stress in asphalt disperses, and low−temperature cracking is prevented.

With the increase in PCM content, the creep stiffness modulus reduced, and the influence rate of different PCM content under all test temperatures was different, indicating that the addition was beneficial for asphalt's low temperature cracking resistance. Only the St of the matrix and SBS−modified asphalt without PCM failed to meet the specification of 300 MPa under all test temperatures, as shown in Figure 7.

The m−value of asphalts with varying PCM concentrations differed marginally from that of asphalt with no PCM added at all test temperatures. At all test temperatures, only the m−values of the matrix and SBS−modified asphalt without PCM were less than 0.3.

There is always a conflict between St and m−values during the evaluation of the low−temperature creep behavior of asphalt. Several researchers have chosen the St/m ratio as a new measurement for analyzing low temperature creep behavior. The St/m ratio is the ratio of the St and m−value at 60 s of test time [46]. A low St/m value is desired. Table 3 indicates that adding PCM to asphalt can improve its resistance to low−temperature cracking.

**Table 3.** The results of St/m at 60 s of test time.

| Temperature/°C | Matrix | Matrix + 4% PCM | Matrix + 8% PCM | SBS | SBS + 4% PCM | SBS + 8% PCM |
|---|---|---|---|---|---|---|
| −6 | 144.3 | 98.2 | 69.6 | 120.1 | 90.7 | 81.7 |
| −12 | 478.7 | 322.7 | 318.9 | 315.8 | 228.9 | 335.2 |
| −18 | 1250.4 | 991.8 | 764.8 | 964.7 | 578.3 | 843.5 |

## 5. Conclusions

PCM is wildly used to regulate the inner temperature of pavement to prevent high− and low−temperature damage. PCM could be classified based on the phase change mechanism, e.g., solid to solid, solid to liquid. In this paper, the effect of solid−solid composite shape−stabilized PCM on the characteristics of matrix and SBS−modified asphalt was investigated. The physical property test, PAV test, Fluorescence Microscope test, FTIR test, and rheological test were employed to confirm the feasibility of directly incorporating solid−solid composite shape−stabilized PCM into the asphalt.

(1) The PCM was physically blended with asphalt in the form of small particles without chemical reaction. Phase separation was obvious. The compatibility between PCM and asphalt was poor;

(2) The addition of PCM could improve the physical properties of asphalt, the PCM concentrations had different influence degrees. The viscosity of SBS-modified asphalt including PCM cannot meet the requirements of the specification. After aging, the high−temperature performance was improved, and low−temperature performance was decreased compared to the asphalt without PCM;

(3) PCM could enhance the high−temperature rutting resistance of the matrix asphalt at a temperature higher than 52 °C at high concentration and reduce the high−temperature rutting resistance of the SBS−modified asphalt to some extent. The high−temperature rutting performance of HMA with PCM compared with HMA without PCM should be evaluated. The procedure of RTOF aging affects the high−temperature rutting resistance of asphalt, especially when PCM concentration is high;

(4) The low−temperature creep behavior and the PG grade of asphalt could be enhanced with the addition of PCM. The St/m ratio could be adopted to evaluate the influence of PCM on the low−temperature cracking resistance of asphalt.

The test results of this study lead to the conclusion that directly incorporating solid−solid composite shape−stabilized PCM into asphalt could alter the physical and rheological properties. The physical property of asphalt is insufficient to assess the impact of PCM, the

rheological test should be adopted for evaluation. It is not a feasible method to directly incorporate PCM into asphalt; hence, the performance of HMA with PCM added should be carefully verified compared with HMA without PCM.

According to this study, the implicating criteria of PCM is that the addition of PCM does not cause deleterious effects on the road performance of the asphalt or asphalt mixture as it enhances the thermoregulation of the asphalt mixture. The addition of PCM cannot increase asphalt pavement performance without incurring the cost of scarifying the performance of the asphalt pavement. The influence of PCM on the physical and rheologdical characteristics of asphalt or asphalt mixture should be investigated and analyzed before application. The desired application approaches of PCM without scarifying the mechanical performance of asphalt or asphalt mixture should be explored.

In this paper, the effect of solid−solid PCM of various concentrations on asphalt was investigated. The method of using solid−solid composite shape−stabilized PCM in asphalt mixture, e.g., PCM blend with aggregates or filler, could be researched further. The effect of solid−solid PCM on the performance of asphalt mixture will be fully investigated in future studies.

**Author Contributions:** Conceptualization, H.Z. (Haisheng Zhao); methodology, S.M.; software, X.W.; validation, H.Z. (Haisheng Zhao), X.W., H.Z. (Huan Zhang) and J.W.; formal analysis, H.Z. (Haisheng Zhao) and Z.L.; investigation, H.Z. (Haisheng Zhao), Z.L. and S.C.; resources, J.G.; data curation, H.Z. (Haisheng Zhao); writing—original draft preparation, H.Z. (Haisheng Zhao); writing—review and editing, S.M.; visualization, H.Z. (Haisheng Zhao), H.Z. (Huan Zhang), C.S., J.W. and S.C.; supervision, S.M.; project administration, J.G. and S.M.; funding acquisition, J.G. All authors have read and agreed to the published version of the manuscript.

**Funding:** This research received no external funding.

**Institutional Review Board Statement:** Not applicable.

**Informed Consent Statement:** Not applicable.

**Data Availability Statement:** Not applicable.

**Acknowledgments:** We thank Zhaodi Yuan and Zhaojie Zhang for their assistance with experiments and valuable discussion.

**Conflicts of Interest:** The authors declare no conflict of interest.

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
