# Peer review of "Effect of Solid-Solid Phase Change Material’s Direct Interaction on Physical and Rheological Properties of Asphalt"

_coatings, doi:10.3390/coatings12050625_

Round 1

Reviewer 1 Report

Dear Authors,

Thank you for your well-structured manuscript, here will be following comments:

  1. please indicate novelty of your study and experimental investigation in abstract, intro and conclusions. Partly it is mentioned in lines 93-95 and 105-108. In general you need to underline if your study is a step forward in this domain. Please also skip using “we, etc.” in description.
  2. the term aging should be spelled “ageing”
  3. can you please explain the necessity of higher temperatures above 60°C in DSR testing?
  4. Your charts have to be elaborated! Very small font and choice of colors plus columns doesn’t make it look like a scientific paper -> figures 1 and 2 and 7
  5. Do you might have more detailed explanation of what actually you see in Figure 3?
  6. Particular reason why your FTIR spectra is different for matrix asphalt from other two measurements in Figure 4a?
  7. please consider to improve quality of Figures 5 and 6 presentation
  8. part of conclusions can be moved to the discussion part, try to make your conclusions – short and clear.

Author Response

I appreciate your suggestions and valuable advice. Based on the suggestions, I modified the manuscripts as follows.

Point 1: please indicate novelty of your study and experimental investigation in abstract, intro and conclusions. Partly it is mentioned in lines 93-95 and 105-108. In general you need to underline if your study is a step forward in this domain. Please also skip using “we, etc.” in description.

Response 1: The novelty of this study was rewritten in the abstract, introduction, and conclusions. The contribution of this study in the application of PCM materials was especially emphasized in the conclusions.

All the expressions about “we, etc” were rewritten.

Point 2: the term aging should be spelled “ageing”

Response 2: All the terms aging in this paper were respelled as “ageing”.

Point 3: can you please explain the necessity of higher temperatures above 60°C in DSR testing?

Response 3: In a previous study, we measured the temperature of the asphalt pavement. When the air temperature was about 36℃, the asphalt pavement temperature of 4 cm below the pavement surface reached 51℃. The high-temperature performance at 60℃ can’t withstand the extreme air temperature influence, so a test temperature that was higher than 60℃ was chosen for DSR tests.

Point 4: Your charts have to be elaborated! Very small font and choice of colors plus columns doesn’t make it look like a scientific paper -> figures 1 and 2 and 7

Response 4: The figures 1, 2, and 7 have been redrawn in a simple form.

Point 5: Do you might have more detailed explanation of what actually you see in Figure 3?

Response 5: The test results and discussion about Figure 3 were rewritten with more description and detailed explanation.

Point 6: Particular reason why your FTIR spectra is different for matrix asphalt from other two measurements in Figure 4a?

Response 6: All the data in Figure 4(a) were thoroughly rechecked, the main reason for the difference between matrix asphalt and the other two measurements is that the baseline used in the FTIR test had small variations. These kinds of variations were not easy to spot during the FTIR test.

Point 7: please consider to improve quality of Figures 5 and 6 presentation

Response 7: The presentations about Figures 5 and 6 were rechecked and modified to provide clear and abundant information.

Point 8: part of conclusions can be moved to the discussion part, try to make your conclusions – short and clear.

Response 8: The conclusions were adjusted, and some parts were removed to exhibit the main conclusions.

Reviewer 2 Report

In order to improve the quality of the paper, please address the following suggestion.

1) p.1, line 29: wildely o widely?

2) p.1, lines 41 and 42: the term "disorders" seem to be inappropriate for a technical paper. Could you replace it with a more appropriate term?

3) p.2, line 48: the terms "disease" seem to be inappropriate for a technical paper. Could you replace it with a more appropriate term?

4) materials and methods: the paragraph is really "short" and the information is too concise. Could it be improved by adding more information about the methodology and characteristics of the materials? e.g., Which is the asphalt concrete mixture? their characteristics? In my opinion, it is important to know how/where the PCM is used. What are the characteristics before and after use? nothing is known about the mixture in general.

5) The research and investigation are well organized but it is difficult to understand (for me) if no information about the asphalt concrete/bitumen characteristics is given. Please add some information about the materials.

6) At the same time some Infos could be added to improve the test description.

Author Response

I appreciate your suggestions and valuable advice. Based on the suggestions, I modified the manuscripts as follows.

Point 1: p.1, line 29: wildely o widely?

Response 1: The right word should be “widely”, and I modified the word.

Point 2: p.1, lines 41 and 42: the term "disorders" seem to be inappropriate for a technical paper. Could you replace it with a more appropriate term?

Response 2: I replaced the word “disorders” with “damages”.

Point 3: p.2, line 48: the terms "disease" seem to be inappropriate for a technical paper. Could you replace it with a more appropriate term?

Response 3: I replaced the word “diseases” with “damages”.

Point 4: materials and methods: the paragraph is really "short" and the information is too concise. Could it be improved by adding more information about the methodology and characteristics of the materials? e.g., Which is the asphalt concrete mixture? their characteristics? In my opinion, it is important to know how/where the PCM is used. What are the characteristics before and after use? nothing is known about the mixture in general.

Response 4: In previous studies, many efforts were focused on the changing in thermoregulation of asphalt or asphalt concrete mixture made by the addition of PCM materials, the influence of PCM material on the characteristics of asphalt was always small attachments. The PCM materials will contact with asphalt when PCM materials are blended into asphalt or asphalt concrete mixture, this study aims to investigate and analyze the influence of solid-solid PCM on matrix and SBS modified asphalts and provide an insight into how the PCM materials affect the characteristics of asphalt under different conditions.

In this study, solid-solid PCM materials with various contents were directly mixed with asphalts, the characteristics of asphalt with and without solid-solid PCM were shown in test results.

The main purpose of this study is to analyze the influence of PCM materials on the characteristics of asphalt, as a result, the influence of solid-solid PCM on asphalt concrete mixture is not included in this study.

Point 5: The research and investigation are well organized but it is difficult to understand (for me) if no information about the asphalt concrete/bitumen characteristics is given. Please add some information about the materials.

Response 5: In this study, matrix asphalt and SBS modified asphalt with 0%, 4wt%, and 8wt% of PCM added were investigated. The physical characteristics of matrix asphalt and SBS modified asphalt before modification were shown in Figure 1. The Physical characteristics of matrix asphalt and SBS modified asphalt after PAC ageing were shown in Figure 2. The morphology of the virgin asphalts were plotted in Figures 3(a) and 3(b). The physical information about the asphalt tested in this study was provided in this form to compare the difference between the virgin and PCM-modified asphalts.

Point 6: At the same time some Infos could be added to improve the test description.

Response 6: Figures 1, 2, and 7 were redrawn in a simple form. Test descriptions were modified with more information for a clear demonstration.

Round 2

Reviewer 1 Report

Your charts have to be elaborated! colors choice -> figures 1 and 2 and 7

Author Response

Point: Your charts have to be elaborated! colors choice -> figures 1 and 2 and 7

Response: I have modified the expression about the results of charts. Figures 1, 2, and 7 were redrawn, and the colors have referred to the website "https://mycolor.space/".

Reviewer 2 Report

The authors addressed the minor comments but did not understand the comment on the materials paragraph improvement.

The goal of the paper is clear. But what the reviewer meant was that to understand the effect of PCMs, information about the materials must be made explicit in the text. The use of different mixture (with different performances) could significantly change the results obtained!
The mixture(s) is a starting point of the study and cannot be ignored.

Author Response

point: 

The authors addressed the minor comments but did not understand the comment on the materials paragraph improvement.

The goal of the paper is clear. But what the reviewer meant was that to understand the effect of PCMs, information about the materials must be made explicit in the text. The use of different mixture (with different performances) could significantly change the results obtained!
The mixture(s) is a starting point of the study and cannot be ignored.

Response: 

Thanks for your suggestions and valuable advice.

The modification of materials and test design were added in the new version, and the test design and material properties would be exhibited more clearly.

I agree that studying asphalt mixtures with PCM is very important. The purpose of the addition of PCM to asphalt mixture is to improve the high-temperature or low-temperature performance of asphalt mixture, the start points of PCM application are good and to solve the damages caused by high or low temperature. Many scholars have made a lot of effort in this area. Most studies were focused on the improvement of the thermal performance of asphalt mixture with the addition of PCM, the negative influence of PCM on the performance of asphalt mixture was obtained small attention.

PCM is a kind of chemical synthetics material that could influence the properties of asphalt and asphalt mixtures. During the studying, not only the improvement of PCM on the thermal performance of asphalt or asphalt mixture should be investigated, but also the potential negative influence o PCM on the physical and rheological properties should be fully analyzed.

The study about PCM should be divided into three phases, the first phase is about the improvement of PCM on the thermal properties of asphalt or asphalt mixture, it is the starting point of PCM exploration. The second phase is to confirm whether the PCM has on positive or negative influence on the properties of asphalt, besides the improvement of thermal properties of asphalt. For the PCM would inevitably contact with asphalt during the application, influence for better or worse would occur during the contact of PCM and asphalt. The influence of PCM on the physical and rheological properties of asphalt should be analyzed before application. The third phase is to investigate the influence of PCM on the performance of asphalt mixture. When the PCM is confirmed to improve the thermal properties of the asphalt mixture and has no negative influence on asphalt, the influence of PCM on the performance of the asphalt mixture except the thermal properties should be investigated. We should confirm that the addition of PCM has no negative influence on the performance of the asphalt mixture.

The three research phases involve the principles of the application of additives to the asphalt mixture, any additives that are added to asphalt mixtures should improve one or more properties, and no property is decreased because of the addition of the additive. The problem about the performance reduction of asphalt mixture couldn’t be covered by the improvement of asphalt mixture performance with the addition of the additive. This paper focused on the influence of PCM on the physical and rheological properties of asphalt based on the principles of PCM application. It is the second phase of the PCM application.

The influence of PCM on the performance of asphalt mixture is the third phase, the influence of PCM on the performance, such as gradation designing, volume indexes, water stability, fatigue performance, indirect tensile strength, dynamic modulus, should be fully investigated. It is a systematic studying topic and is not involved in the same paper.
